# Three-year mortality in cryptococcal meningitis: Hyperglycemia predict unfavorable outcome

Sheng-Ta Tsai[ORCID]¹, Fu-Yu Lin[ORCID]¹*, Pei-Shan Chen², Hsiu-Yin Chiang², Chin-Chi Kuo²,³

1 Department of Neurology, China Medical University Hospital and College of Medicine, China Medical University, Taichung, Taiwan, 2 Big Data Center, China Medical University Hospital and College of Medicine, China Medical University, Taichung, Taiwan, 3 Division of Nephrology, Department of Internal Medicine, China Medical University Hospital and College of Medicine, China Medical University, Taichung, Taiwan

* linfuyu95@gmail.com

## Abstract

Existing evidence revealed grave prognosis for cryptococcal meningitis (CM), particularly its short-term mortality. However, its long-term survival and prognostic factors remained unknown. This study investigated 3-year mortality and analyzed its predictive factors in patients with CM. This retrospective cohort study with 83 cerebrospinal fluid culture-confirmed CM patients was conducted at China Medical University Hospital from 2003 to 2016. The 3-year mortality rate in patients with CM was 54% (45 deaths among 83 patients). Advanced age, human immunodeficiency virus (HIV) seronegative state, low Glasgow Coma Scale score on admission, decreased hemoglobin and hyperglycemia on diagnosis were associated with 3-year mortality. After multivariate adjustment in the Cox proportional hazard model, only severe hyperglycemia (serum glucose ≥200 mg/dL) on diagnosis could predict 3-year mortality.

## Introduction

*Cryptococcus* species, most commonly causing cryptococcal meningitis (CM), was estimated to cause over 180,000 deaths per year globally. The mortality of CM with human immunodeficiency virus (HIV) infection in low-income countries might exceed 70% [1, 2]. Although CM was considered rare among non-immunosuppressed hosts, the prognosis remained poor in developed countries [3]. A systematic review focusing on the outcomes of adult patients with CM over more than 3 months showed that 1-year mortality could reach up to 78% in HIV patients with CM and 42% in non-HIV patients with CM [4]. Previous studies [4] suggested older age, altered neurological status, and high fungal burden were risk factors for 3-month to 1-year mortality in patient with CM. Lacking of antiretroviral therapy (ART) and low CD4 cell count at induction predicted poor prognosis specifically for HIV patients with CM. By contrast, delayed diagnosis and coexistence of underlying comorbidities were related to high mortality in non-HIV patients with CM [4–6]. Although the in-hospital mortality rate was more than 50% in sub-Saharan Africa [7, 8], a report showed that the 5-year survival of HIV patients

**Data Availability Statement:** All relevant data are within the paper and its Supporting Information files.

**Funding:** This study was supported by grants from the Ministry of Science and Technology, Taiwan

(Grant number: MOST 108-2314-B-039-038-MY3 and MOST 109-2321-B-468- 001) and from CMUH (DMR-109-130). This study was not sponsored by industry. The funders had no role in study design, data collection and analysis, decision to publish, or preparation of the manuscript.

**Competing interests:** The authors have declared that no competing interests exist.

with CM and a survival time of at least 6-months was excellent, indicating that long-term survival might be possible [9]. Current therapeutic guideline recommended an at least one-year pharmacological treatment program for HIV patients with CM [10]. However, a survey of prognostic factors for more than 1 year outcomes was still lacking. Therefore, we conducted this retrospective CM cohort study to investigate 3-yeair mortality in patients with CM and the associated predictive factors.

## Methods

### Study population

This retrospective cohort study was conducted using data from China Medical University Hospital's Clinical Research Data Repository (CMUH-CRDR), which contained the medical records of 2,660,472 patients who sought treatment from January 1, 2003 to December 31, 2016 [11, 12]. From the CMUH-CRDR data, we identified 11,910 microbiology cultures for *Cryptococcus* species and obtained 181 positive cultures from the cerebrospinal fluid (CSF) of 83 patients. We included these 83 patients who had isolation of *Cryptococcus* species from CSF as the study population. The index date was the date when the CSF cultures were found to be positive for *Cryptococcus* species. The Research Ethical Committee/Institutional Review Board of China Medical University Hospital approved this study (CMUH105-REC3-068).

### Covariates and outcome of interest

Electronic medical records (EMR) data of comorbidities, medications, and biochemical profiles were obtained from the CMUH–CRDR, except for the data for lumbar puncture pressure and symptoms that were obtained after an EMR review by a neurologist. Covariates evaluated in this study included the following: age, sex, and comorbidities within 1 year prior to the index date (diabetes and hypertension were defined based on the relevant International Classification of Diseases [ICD] codes and the medications received; other comorbidities were defined based on the relevant ICD diagnosis codes; S1 Table). Covariates related to CM included the following: symptoms; Glasgow Coma Scale (GCS) on admission; records of lumbar puncture; corticosteroid use within 90 days prior to the index date (including cortisone acetate, dexamethasone, prednisolone, and methylprednisolone); CSF cryptococcal antigen; India ink staining; CSF profiles of white blood cells, red blood cells, total protein, and glucose that were obtained closest to and within ±30 days of the index date; medical treatment during hospitalization (medications, including amphotericin B, liposomal amphotericin B, flucytosine, fluconazole, and itraconazole), and surgical management for intracranial hypertension. Blood cultures and serum biochemical profiles that were obtained closest to and within 30 days prior to the index date were acquired. Random non-fasting serum glucose cutoff levels of 140 mg/dL (7.8 mmol/L) and 200 mg/dL (11.1 mmol/L) were used to define hyperglycemia and severe hyperglycemia, respectively [10]. The primary outcome of interest was 3-year all-cause mortality following the index date, where the mortality was ascertained using the data from the National Death Registry. We performed multiple imputations using an iterative Markov chain Monte Carlo procedure with 20 imputations and 100 iterations [13]. We used the data from the multiple imputations in the subsequent multivariate analysis.

### Statistical analysis

Continuous variables were presented as medians and interquartile ranges and were compared using the Wilcoxon rank-sum test. Categorical variables were expressed as frequency and percentage and were compared using the chi-square test or Fisher's exact test. Based on

pathophysiological rationales, 4 most relevant exposures for mortality were selected. To determine the association between exposure and 3-year all-cause mortality, we analyzed each exposure in univariate Cox proportional hazard model with age as the time scale followed by multivariate adjustment. All statistical analyses were performed using R version 3.2.3 (The R Foundation for Statistical Computing, Vienna, Austria). The 2-sided statistical significance level was set at α = 0.05.

## Results

In this study, a total of 83 patients were included in the CM cohort (Table 1). The mean age was 49 years with a male predominance (77.1%). The HIV-infected and HIV-seronegative patient percentage were 39.8% (33 of 83 patients) and 60.2% (50 of 83 patients), respectively. The percentage of HIV patients who have been accepted antiretroviral therapy prior to CM diagnosis was 33.3% (11 of 33 patients). Patients without comorbidities including HIV infection, organ transplantation, malignancy, diabetes mellitus, autoimmune disease, liver cirrhosis or chronic kidney diseases were 25.3% (21 of 83 patients). Fever, headache, and altered mental status were common symptoms. The pathogen percentage of *Cryptococcus* neoformans and *Cryptococcus* gattii was 98.8% (82 of 83 patients) and 1.2% (1 of 83 patients), respectively (S2 Table). Among study population, 81.9% (68 of 83) received an amphotericin B-based regimen as induction therapy, 91.6% (76 of 83) patients received fluconazole, and 1.2% (1 of 83) received itraconazole for consolidation and maintenance. All the patients had consulted an infection specialist according to the hospital infection control policy. Regarding surgical intervention for increased intracranial pressure control, 30.1% (25 of 83 patients) received surgical shunt.

The 3-year mortality was 54.2% (45 deaths in 83 patients). Factors that were significantly frequent in the 3-year all-cause mortality group were as follows: advanced age, HIV-seronegative status, liver cirrhosis, low GCS score on admission, decreased hemoglobin on diagnosis, and hyperglycemia on diagnosis (P<0.01, Table 1). Then we did the univariate and multivariate Cox proportional hazards analysis for these four important factors (Table 2). Only severe hyperglycemia (serum glucose ≥ 200 mg/dL) was significantly associated with 3-year mortality (adjusted HR = 7.167, 95% CI: 1.4–36.7) in multivariate Cox proportional hazards analysis.

## Discussion

The present study showed that the 3-year mortality rate in the cohort with CM was 54.2% (45 deaths in 83 patients). Previous studies showed high short-term mortality in patients with CM [1–4]. A study in Uganda found that the 5-year mortality rate of HIV-infected patients with symptomatic CM was 58% [9]. Our study results revealed that the 3-year mortality of HIV-infected patients with symptomatic CM was 30.3% (10 deaths in 33 patients), which is lower than that in the aforementioned study cohort. Severe hyperglycemia on diagnosis, defined as serum glucose ≥200mg/dL, predicted 3-year mortality among patients with CM.

In earlier studies, the treatment outcomes between studies varied according to the local availability of medical resources, especially for the accessibility of adequate anti-fungal agents [14]. The 1-year mortality of HIV-associated CM in low- and middle-income countries, Europe (including Russia) and North America was estimated to be approximately 70%, 40%, 30% and 20%, respectively [1]. In this study, we consecutively recruited HIV-infected and HIV-seronegative patients. Most patients, regardless of the HIV status, received amphotericin B-based anti-fungal regimens. In our cohort, 3-year mortality in HIV-infected patients with CM was 30.3% (10 deaths in 33 patients), which was consistent with those in high-income countries. By contrast, 3-year mortality in HIV-seronegative patients with CM was 70% (35

**Table 1. Demographic and clinical characteristics of patients with cryptococcal meningitis during 2003–2016 at China Medical University Hospital.**

| Variables[a] | Overall | Died within 3 years | Alive within 3 years | P-value[b] |
|---|---|---|---|---|
| N | N = 83 | N = 45 | N = 38 | |
| Age at index date[c] (year) | 49 (34.5, 64.5) | 57 (45, 71) | 37 (30, 49) | <0.001 |
| Male | 64 (77.1) | 32 (71.1) | 32 (84.2) | 0.249 |
| **Comorbidities[d]** | | | | |
| Human immunodeficiency virus | 33 (39.8) | 10 (22.2) | 23 (60.5) | 0.001 |
| Organ transplantation | 2 (2.4) | 2 (4.4) | 0 (0.0) | 0.498 |
| Malignancy | 8 (9.6) | 7 (15.6) | 1 (2.6) | 0.065 |
| Diabetes mellitus | 9 (10.8) | 7 (15.6) | 2 (5.3) | 0.170 |
| Autoimmune disease | 5 (6) | 3 (6.7) | 2 (5.3) | 1.000 |
| Liver cirrhosis | 10 (12) | 10 (22.2) | 0 (0.0) | 0.002 |
| Chronic kidney disease | 5 (6) | 4 (8.9) | 1 (2.6) | 0.369 |
| None of above | 21 (25.3) | 10 (22.2) | 11 (28.9) | 0.654 |
| **Corticosteroid history within 90 days prior to the index date** | | | | |
| Cortisone acetate | 2 (2.4) | 2 (4.4) | 0 (0.0) | 0.498 |
| Dexamethasone | 5 (6) | 4 (8.9) | 1 (2.6) | 0.369 |
| Prednisolone | 8 (9.6) | 6 (13.3) | 2 (5.3) | 0.279 |
| **Antiretroviral therapy within 1 year prior to the index date** | 11 (13.3) | 6 (16.3) | 5 (13.2) | 1.000 |
| **Symptoms, n (%)** | | | | |
| Fever | 41 (50) | 27 (60.0) | 14 (36.8) | 0.060 |
| Headache | 48 (58.5) | 22 (48.9) | 26 (68.4) | 0.116 |
| Neck stiffness | 7 (8.5) | 2 (4.4) | 5 (13.2) | 0.238 |
| Altered Mental Status | 33 (40.2) | 23 (51.1) | 10 (26.3) | 0.038 |
| Visual symptoms | 9 (11) | 3 (6.7) | 6 (15.8) | 0.289 |
| Auditory symptoms | 8 (9.8) | 4 (8.9) | 4 (10.5) | 1.000 |
| Seizures | 9 (11) | 4 (8.9) | 5 (13.2) | 0.726 |
| Glasgow Coma Scale within—/+3 days of the index date | 15 (10, 15) | 10 (8, 15) | 15 (14.2, 15) | 0.006 |
| **Lumbar puncture during hospitalization** | | | | |
| First open pressure (cmH$_2$O) | 24 (16, 29.2) | 23 (16, 28) | 25.7 (20, 37.5) | 0.151 |
| **Profiles of CSF within -/+ 30 days of the index date** | | | | |
| Positive *Cryptococcal* antigen | 73 (88.0) | 39 (86.7) | 34 (89.5) | 0.748 |
| Positive India Ink | 66 (79.5) | 37 (82.2) | 29 (76.3) | 0.696 |
| White blood cell (/ul) | 47 (5, 156) | 63 (7, 150) | 24.5 (5, 158) | 0.568 |
| Red blood cell (/ul) | 23 (3, 155) | 26 (8, 180) | 14 (2, 70.5) | 0.109 |
| Total protein (mg/dL) | 92.8 (53.5, 189.1) | 104.5 (54.5, 225.8) | 81.5 (52.0, 151.0) | 0.162 |
| CSF glucose (mg/dL) | 39 (16, 53.5) | 41 (16.0, 59.0) | 35 (16.0, 51.5) | 0.667 |
| **Positive blood culture that grew *Cryptococci*** | 29 (34.9) | 15 (33.3) | 14 (36.8) | 0.918 |
| **Serum biochemical profiles within 30 days prior to the index date** | | | | |
| Positive *Cryptococcal* antigen | 61 (73.5) | 33 (73.3) | 28 (73.7) | 1.000 |
| Serum glucose (mg/dL) | 122.5 (109.8, 166.5) | 154.5 (115.2, 195) | 113 (106.2, 127.8) | < 0.001 |
| <140 mg/dL | 50 (60.2) | 17 (37.8) | 33 (86.8) | < 0.001 |
| 140–199 mg/dL | 17 (20.5) | 13 (28.9) | 4 (10.5) | |
| ≧200 mg/dL | 15 (18.1) | 14 (31.1) | 1 (2.6) | |
| ESR (mm/hr) | 31 (16, 61) | 36 (15.5, 59) | 23.5 (16.5, 60.5) | 0.769 |
| White blood cell (1x10$^3$/ul) | 8 (5.8, 10.9) | 8.6 (5.8, 13.3) | 7 (5.8, 10.1) | 0.283 |
| Creatinine (mg/dL) | 0.9 (0.7, 1) | 0.9 (0.7, 1.1) | 0.8 (0.7, 1) | 0.475 |
| Alanine aminotransferase (ALT) (IU/L) | 26.5 (18.2, 50.8) | 32.5 (23, 58) | 22 (14, 44) | 0.033 |
| Hemoglobin (g/dL) | 11.9 (10.1, 13.3) | 10.7 (9.4, 12.3) | 12.6 (12, 13.8) | < 0.001 |

*(Continued)*

**Table 1.** (Continued)

| Variables[a] | Overall | Died within 3 years | Alive within 3 years | P-value[b] |
|---|---|---|---|---|
| CD4 / mm$^3$ | 34.5 (19.5, 67.5) | 30.5 (15.8, 64.5) | 34.5 (23.2, 75.2) | 0.496 |
| **Medical treatment during hospitalization** | | | | |
| Amphotericin B or Liposomal Amphotericin B | 68 (81.9) | 37 (82.2) | 31 (81.6) | 1.000 |
| Amphotericin B | 64 (77.1) | 34 (75.6) | 30 (78.9) | 0.917 |
| Liposomal Amphotericin B | 20 (24.1) | 11 (24.4) | 9 (23.7) | 1.000 |
| Flucytosine | 60 (72.3) | 33 (73.3) | 27 (71.1) | 1.000 |
| Fluconazole | 76 (91.6) | 40 (88.9) | 36 (94.7) | 0.445 |
| Itraconazole | 1 (1.2) | 0 (0.0) | 1 (2.6) | 0.458 |
| Surgical shunt | 25 (30.1) | 11 (24.4) | 14 (36.8) | 0.324 |

Abbreviations: CSF, cerebrospinal fluid; ESR, erythrocyte sedimentation rate; IQR, interquartile range.

a. Categorical variables are presented as frequency (%) and continuous variables are presented as median (IQR), if not otherwise specified.

b. P-values are calculated by Kruskal-Wallis test for continuous variables and Chi-square test (or Fisher's exact test as appropriate) for categorical variables.

c. Index date was the date of positive cerebrospinal fluid culture that grew Cryptococcal species.

d. Comorbidities were defined by the ICD diagnosis codes corresponding to each disease (S1 Table) that were recorded within one year prior to the index date.

deaths in 50 patients) (S3 Table). The high mortality in HIV-seronegative patients with CM in our cohort may be relate to advanced age (59 years old v.s. 34 years old) and the presence of other comorbidities. In a similar tertiary medical center in South Korea, no difference in mortality was found between HIV-infected and HIV-seronegative patients; corresponding to HIV-infected (48 years) and HIV-seronegative (46.5 years) patients in their cohort had similar age [15]. In a retrospective survey conducted in 26 centers of 11 countries, only the coexistence of malignancy was associated with poor outcomes [5].

The use of effective antiretroviral therapy for HIV-infected patients contributed to the decline of cryptococcosis incidence [16]. In our cohort, the percentage of previous use of antiretroviral therapy in HIV-infected CM patients was 33.3% (11 of 33 patients). For HIV-seronegative patients, the host immune status was heterogeneous. Our study identified 25.3% (21 of 83 patients) otherwise "normal" patients without comorbidities including HIV infection, organ transplantation, malignancy, diabetes mellitus, autoimmune disease, liver cirrhosis or chronic kidney disease. This phenotypically "normal" group probably represented the congruence of subclinical innate or acquired immunodeficiencies [17].

Notably, in our cohort, 3-year mortality in patients with CM and liver cirrhosis was 100% (10 deaths in 10 patients). The relationship of liver cirrhosis with cryptococcal disease has

**Table 2. Univariate and multivariate Cox proportional hazards analysis for the outcome of 3-year mortality.**

| Variables | Univariate | | Multivariate | |
|---|---|---|---|---|
| | Crude HR | P-value | Adjusted HR | P-value |
| | (95% CI) | | (95% CI) | |
| **Glasgow Coma Scale** | 0.947 (0.80, 1.12) | 0.513 | 0.992 (0.81,1.22) | 0.937 |
| **Human Immunodeficiency Virus** | 0.324 (0.10, 1.06) | 0.063 | 0.353 (0.09,1.44) | 0.144 |
| **Hemoglobin (g/dL)** | 0.804 (0.66, 0.97) | 0.024* | 0.920 (0.73,1.17) | 0.485 |
| **140≦Glucose<200 (Ref. Glucose<140)** | 2.247 (0.83, 6.07) | 0.109 | 2.255 (0.70,7.23) | 0.169 |
| **Glucose≧200 (Ref. Glucose<140)** | 8.586 (2.06, 35.7) | 0.004* | 7.167 (1.4,36.7) | 0.019* |

Abbreviations: HR, hazard ratio.

*: P < 0.05

been studied [18]. A comparative study of the outcomes of cryptococcal disease between patients with advanced liver disease and HIV infection revealed that end-stage liver disease was a strong predictor of early mortality [19]. Our data was consistence to the aforementioned findings, but the sample size was too small for further analysis. In addition, our data showed that decreased hemoglobin on diagnosis was associated with 3-year mortality, but the effect disappeared after multivariate correction. An earlier study revealed that HIV-infected patients with CM and moderate to severe anemia at baseline had an increased mortality risk in the short term [20]. Although the possible mechanism of the association between anemia and mortality seemed to be driven by factors presented at diagnosis rather than treatment toxicity, still remained unknown.

Our analysis demonstrated that severe hyperglycemia (serum glucose $\geq$200 mg/dL) on diagnosis was associated with 3-year mortality. This relation remained significant in both univariate and multivariate Cox proportional hazards model with age as the time scale. By contrast, diabetes mellitus in our cohort was not associated with 3-year mortality. In a study of critically ill patients in the intensive care unit, stress hyperglycemia could be considered a biomarker of disease severity and was strongly associated with mortality [21]. Hyperglycemia has also been recognized as an indicator of poor clinical outcomes in patients with acute stroke and traumatic brain injury, resulting from oxygen stress, lactic acidosis, inflammation and disruption of the blood-brain barrier [22, 23]. Laboratory data revealed that the hyperglycemic state caused macrophage dysfunction and the inhibition of tumor necrosis factor production [24–26]. However, the association of diabetes mellitus and outcomes of patients with CM was controversial [26–30]. Our results suggested that severe hyperglycemia on diagnosis, rather than the comorbidity of diabetes mellitus, could predict 3-year mortality in patients with CM.

The strength of this study was long-term mortality analysis. We conducted 3-year follow-up compared with 1-year follow-up period in most studies because we were capable of linking the local clinical database with National Death Registry. Our results were obtained from a single medical center, which had controlled care quality, and enrolled both HIV-infected and HIV-seronegative patients. However, this study had certain limitations. First, the retrospective, single-center design and small sample size might have underestimated some important predictive factors. Second, we did not perform detailed microbiological investigation, such as serotypes. Lastly, we did not evaluate 3-year neurological deficits or functional outcomes.

In summary, we retrospectively analyzed 83 patients with CSF culture-confirmed CM between 2003 and 2016 at a single medical center in central Taiwan. The 3-year mortality was 54%. After risk factors adjustment, severe hyperglycemia (serum glucose $\geq$200 mg/dL) on diagnosis predicted 3-year mortality. Further prospective study needs to be conducted for precise outcome analysis of this disease.

## Supporting information

**S1 Table. ICD diagnosis codes and medication for comorbidities used in this study.**
(DOCX)

**S2 Table. Cryptococcal species that grew in the cerebrospinal fluid.**
(DOCX)

**S3 Table. Demographic and clinical characteristics of patients with cryptococcal meningitis during 2003–2016 at China Medical University.**
(DOCX)

## Acknowledgments

We appreciate the data exploration, statistical analysis, manuscript preparation, and the support of the iHi Clinical Research Platform from the Big Data Center of CMUH. We would like to thank the Health and Welfare Data Science Center (HWDC), Ministry of Health Welfare, and Health Data Science Center, China Medical University Hospital for providing administrative and technical support. We also thank Enago (www.enago.tw) for the English language review.

## Author Contributions

**Conceptualization:** Sheng-Ta Tsai, Fu-Yu Lin.

**Data curation:** Pei-Shan Chen, Hsiu-Yin Chiang.

**Formal analysis:** Pei-Shan Chen, Hsiu-Yin Chiang.

**Methodology:** Pei-Shan Chen, Hsiu-Yin Chiang.

**Project administration:** Chin-Chi Kuo.

**Resources:** Chin-Chi Kuo.

**Software:** Chin-Chi Kuo.

**Supervision:** Chin-Chi Kuo.

**Validation:** Chin-Chi Kuo.

**Writing – original draft:** Sheng-Ta Tsai.

**Writing – review & editing:** Fu-Yu Lin.

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
