## [Decision Letter · Decision Letter 0]

1 Feb 2021

PONE-D-20-37177

Three-year Mortality in Cryptococcal Meningitis: Hyperglycemia Predict Unfavorable Outcome

PLOS ONE

Dear Dr. Lin,

Thank you for submitting your manuscript to PLOS ONE. After careful consideration, we feel that it has merit but does not fully meet PLOS ONE’s publication criteria as it currently stands. Therefore, we invite you to submit a revised version of the manuscript that addresses the points raised during the review process.

We look forward to receiving your revised manuscript.

Kind regards,

Karen L. Wozniak, PhD

Academic Editor

PLOS ONE

Journal Requirements:

3.Thank you for stating the following in the Acknowledgments Section of your manuscript:

"This study is supported by big data center in china medical university hospital."

 "No. The funders had no role in study design, data collection and analysis, decision to publish, or preparation of the manuscript"

Reviewers' comments:

Reviewer's Responses to Questions

**Comments to the Author**

1. Is the manuscript technically sound, and do the data support the conclusions?

Reviewer #1: Partly

Reviewer #2: Yes

2. Has the statistical analysis been performed appropriately and rigorously? 

Reviewer #1: Yes

Reviewer #2: Yes

3. Have the authors made all data underlying the findings in their manuscript fully available?

Reviewer #1: Yes

Reviewer #2: Yes

4. Is the manuscript presented in an intelligible fashion and written in standard English?

Reviewer #1: Yes

Reviewer #2: Yes

5. Review Comments to the Author

Reviewer #1: In this manuscript, Tsai et. al. evaluated prognostic factors for long-term survival for cryptococcal meningitis. The authors evaluated 3-year mortality in a retrospective cohort study with 83 confirmed CM patients in China Medical University Hospital. Their analysis showed that advanced age, human immunodeficiency virus (HIV) seronegative state, low Glasgow Coma Scale score on admission, decreased hemoglobin and hyperglycemia on diagnosis were associated with 3-year mortality. After multivariate adjustment, only severe hyperglycemia was associated with 3-year mortality. The reviewer has several concerns about the study:

1. The introduction can be strengthened if the authors discuss what is previously known about predictive factors of short and long-term mortality and why a new long-term mortality analysis is needed.

2. it is unclear whether prognostic factors predict long-term mortality analysis also predict short-term mortality. The authors should discuss if such differences exist.

3. Can the authors analyze or discuss if similar prognostic factors predict mortality for CM patients that are immune-suppressed (due to HIV/treatments) or immune-competent ?

Reviewer #2: The authors of this manuscript performed a retrospective study of 83 patients with confirmed cryptococcal meningitis (CM). One overall conclusion is that severe hyperglycemia is a better predictor of 3-yr mortality in CM patients than comorbidity of diabetes mellitus. However, one important point that deserves more detailed discussion is the lower mortality rate in CM patients with HIV when compared to CM patients without HIV infection. The manuscript would benefit from a more detailed analysis of the differences between these two groups of patients. If the data is available, it would be very interesting to know the differences in age distribution, presence of comorbidities and some discussion if HIV treatment itself had any effect on CM mortality in CM patients with HIV infection. While the study is limited in its sample size, the authors have performed sufficient comparison and statistical analysis to reach their conclusions.

6. PLOS authors have the option to publish the peer review history of their article (what does this mean?). If published, this will include your full peer review and any attached files.

Reviewer #1: No

Reviewer #2: **Yes: **Avishek Mitra

---

## [Author Response · Author response to Decision Letter 0]

10 Apr 2021

We replied to two reviewers in detail and made four more tables to strengthen the result. We wrote our response in the “response to reviewers” file. Thanks for your insightful comments.

---

## [Editor Report · Decision Letter 1]

3 May 2021

Three-year Mortality in Cryptococcal Meningitis: Hyperglycemia Predict Unfavorable Outcome

PONE-D-20-37177R1

Dear Dr. Lin,

We’re pleased to inform you that your manuscript has been judged scientifically suitable for publication and will be formally accepted for publication once it meets all outstanding technical requirements.

Kind regards,

Karen L. Wozniak, PhD

Academic Editor

PLOS ONE

---

## [Editor Report · Acceptance letter]

21 May 2021

PONE-D-20-37177R1 

Three-year Mortality in Cryptococcal Meningitis: Hyperglycemia Predict Unfavorable Outcome 

Dear Dr. Lin:

I'm pleased to inform you that your manuscript has been deemed suitable for publication in PLOS ONE. Congratulations! Your manuscript is now with our production department. 

Kind regards, 

on behalf of

Dr. Karen L. Wozniak 

Academic Editor

PLOS ONE